# Using natural vegetation succession to evaluate how natural restoration proceeds under different climate in Yunnan, Southwest China

**Weifeng Gui**[1,2], **Qingzhong Wen**[1]*, **Wenyuan Dong**[2], **Xue Ran**[3], **Xiaosong Yang**[1], **Guangqi Zou**[1], **Dechang Kong**[1]

**1** Yunnan Institute of Forest Inventory and Planning, Kunming, China, **2** Southwest forestry university, Kunming, China, **3** Yunnan College of Tourism Vocation, Kunming, China

* qingzhongwen@outlook.com

## Abstract

Currently, natural restoration has been widely proposed as the primary method of ecological restoration and has been studied for a long time. However, research on how to quantify the progress of natural restoration in different climate conditions, especially using long-term succession monitoring data combined with habitat quality data across various succession stages, has been scarce. Our study aims to address this issue in Yunnan, southwest China. To quantify the progress of natural restoration under different climates in Yunnan, we introduced an index, the Natural Succession Index. Utilizing topography and meteorological data, we divided the study area into different climate sub-areas using the Two-stage clustering algorithm. We then combined 1703 sets of 30-year succession monitoring data, each with six observations taken at five-year intervals from 1987 to 2017, with habitat quality data from different succession stages (grassland, shrub, and forest) to quantify the Natural Succession Index. Yunnan province was divided into 14 sub-areas, namely C(I to II), M(I to III), W(I to IV), and H(I to IV), each possessing a unique environment. The indices in each sub-area were calculated, with the results showing a specific order: H-I (0.7812) > H-IV (0.7739) > W-I (0.6498) > M-III (0.6356) > H-III (0.6316) > M-II (0.5735) > W-III (0.5644) > W-IV (0.5571) > C-II (0.4778) > W-II (0.3980) > M-I (0.3624) > H-II (0.3375) > C-I (0.2943). The times for natural succession to reach the forest stage vary from 5 to 19 years, which aligns with the order of indices. The stand volumes of vegetation in the forest stage range from 5 m³ to 110 m³, with a higher Natural Succession Index value corresponding to a higher stand volume of vegetation. In the future, the index could be utilized to reallocate investments in natural restoration projects for better returns. Constant vigilance is required in the first five years following the implementation of restoration actions to avoid failure due to calculation errors.

**Data availability statement:** All relevant data are within the paper and its Supporting Information files.

**Funding:** The author(s) declare that financial support was received for the research, authorship, and/or publication of this article. This study was funded by a grant from the Protection and Restoration of Ecological Space Research Project in Yunnan Province (2020103122 to QZW and WFG).

**Competing interests:** The authors have declared that no competing interests exist.

## Introduction

In 2015, the United Nations introduced 17 Sustainable Development Goals (SDGs) aimed at addressing the threats posed by ecological degradation and overdevelopment through sustainable approaches. Similarly, in 2012, China released a policy on conservation culture, aligning with the same motivations as the UN. Ecological restoration be regarded as a key way to mitigate ecological degradation [1].

Natural restoration, also often referred to as natural regeneration or unassisted restoration, is the process of recovery that occurs without active human intervention [2]. Many scholars are increasingly interested in natural restoration to mitigate ecological degradation, considering it to be both effective and sustainable [3]. Vegetation resulting from natural restoration is often equal to or even superior in structure and function to that produced by directed succession [4–6].

Natural vegetation succession is regarded as the key point in natural restoration, and many scholars consider that understanding the succession processes is necessary for successful restoration [7–10]. Prach et al. mentioned a challenge for ecologists which is to enhance the integration of scientific knowledge of natural vegetation succession into restoration programs and to ensure effective implementation of theoretical and practical information [11].

Numerous studies have explored the natural vegetation succession in natural restoration. Duan et al. conducted a study involves natural vegetation succession of degraded lowland in South China [12]..Alena et al. surveyed the species richness, the direction of natural succession and the community composition of 58 fields which been abandoned for three decades and fully recovered by natural succession [13]. Sawtschuk et al. regarded natural succession as a restoration tool for degraded maritime cliff-top vegetation in Brittany [14]. Mota et al. spent 13 years to reveal how natural primary succession happened in gypsum quarries in Iberian by natural restoraton [15]. However, almost researches relative to natural succession for restoration are focuses on the changes of community composition and species. Prach et al. have mentioned that studies of natural vegetation succession for natural restoration related to quantitatively compare in a larger scale are rare [11]. And today, there are still rarely studies conducted in a large scale, let alone a quantification study.

The effectiveness of natural restoration is heavily depend on habitat quality because of the key role of natural vegetation succession in natural restoration, as highlighted in numerous studies. For instance, succession in the humid and rainy southern subtropical regions is relatively easy and rapid [16,17]. In contrast, it is hard and slow in the alpine regions on the southeastern edge of the Tibetan Plateau [17]. The landform of Yunnan province is very complex and varied because of the collide between the Eurasian plate and Indian Ocean plate [18]. The collide formed several geographical units in Yunnan. Different geographical units formed each specific climate conditions. The climate conditions types include northern temperate climate, subtropical climate and temperate climate [19]. Above all, how to confirm natural restoration works for different climate conditions is promising for Yunnan. In addition, this study is also promising for other areas with various climate conditions because natural restoration are not only implemented in Yunnan province, but also be wildly implemented in whole world [20–24].

In this study, we proposed a hypothesis that nature restoration would not work in a same effectiveness under different climate conditions, because the restoration relies on natural vegetation succession, and we could intuitively show or compare the effectiveness in different climate conditions. For this, we introduced the Natural Succession Index(NSI), a metric designed to quantify the influence of the habitat on the difficulty of natural vegetation succession in a given region. we utilized 1331 succession monitoring records from 1987 to 2017, recent meteorological data from the past decade, and topographic data to analyze the Natural

Succession Index(NSI) across various climatic sub-areas. Our goal was to determine whether natural restoration is effective throughout Yunnan and to understand the mechanisms by which it operates. Ultimately, we aimed to develop an innovative approach to enhance the success rate of restoration at a regional scale by employing the Natural Succession Index(NSI).

## Materials and methods

### Study site

Yunnan Province, located in southwest China, spans an area of 394,100 km², accounting for approximately 4% of China's total land area. The terrain gradually descends from northwest to southeast, featuring a wide range of elevations. The majority of mountains in Yunnan run from west to north due to the collision between the Eurasian and Indian Ocean plates. The climate is diverse, encompassing six distinct types: the northern tropical zone, the southern subtropical zone, the middle subtropical zone, the northern subtropical zone, the temperate zone, and the plateau climate zone. This climatic diversity greatly contributes to the rich variety of habitats in Yunnan.

### Materials and monitor system

In this study, the study area has been divided into climate zone based on 1391 towns of Yunnan province. Every towns were regarded as one sample, and possessed one set of environmental factors. The environmental factors include climate and geography. The climate factors included annual average temperature, precipitation, relative humidity, and annual accumulated temperature, all collected from weather stations. The geographic factors were represented by altitude, obtained from the Shuttle Radar Topographic Mission (SRTM) Digital Terrain Model (DTM) with a 90 m resolution, available from www.srtm.usgs.gov. These factors (annual precipitation, annual average temperature, altitude, relative humidity, and annual accumulated temperature) were selected for division due to their significant roles in the vegetation succession [25,26].

For evaluating how natural restoration proceeds in different environments access the natural vegetation succession, we analyzed 1331 succession monitoring records to calculate the Natural Succession Index(NSI) and reveal the community characteristics in vegetation succession. These records were extracted from a long-term forest resource monitoring

system which systematically sampled the entire province using a 6km × 8km grid system. The system has been in place since 1987, with data collected every five years until 2017. The monitoring system is a major forest monitoring project in China, aimed at clarifying the status of forest resources. Each monitoring sample was a square with a 28.28m side. The layer of forest, shrub and herb were observed respectively. The observation data of forest were collected by the square with a 28.28m side. The observation data of shrub were collected by three square with a 4m side inside of the forest square, and the shrub square evenly distributed in one diagonal line of forest square. The observation data of grass were collected by five square with a 1m side inside of the forest square, and the grass square distributed at the four corners and center of the forest square.

The data encompasses 85 research factors, including land type information, terrain details (e.g., altitude, slope, aspect), community characteristics (e.g., dominant species in each life form, cover degree, height, DBH, origin), and disturbance information (e.g., logging, grazing, cutting, types of disasters).

These samples share the following characteristics: 1) in 1987, the land type was grassland dominated by natural species; 2) few seed trees and saplings were present; 3) the disturbance information of grazing and cutting was recorded on these samples at 1987, but disappeared

in last 25 years. 4) none of the samples experienced significant natural disasters or human interventions in last 25 years; and 5) the samples were protected by our national protection action(artificial patrol protection), recovered fully by natural vegetation succession. The distribution of these samples is detailed in Fig 1.

## Climate division

A two-stage clustering algorithm was employed for subdivision. R was used to conduct the analysis. GGtree package was used to perform cluster analysis step by step. Arcgis 10.8.2 was used to exhibit the cluster analysis results. The environmental data in every sample was extracted by Arcgis 10.8.2. The Shuttle Radar Topographic Mission (SRTM) Digital Terrain Model (DTM) with a 90 m resolution, was used to generate the topographic data in format TIF. Climate data was collected from weather stations point to point. The annual accumulated temperature (the aggregate sum of daily mean temperatures exceeding 0°C, accumulated over a specified temporal interval) was considered the primary factor for the initial division, because there is study indicated that while precipitation and temperature are crucial to vegetation growth, the impact of precipitation diminishes in areas with abundant rainfall, where accumulated temperature becomes the primary influence on vegetation [27]. Annual precipitation, annual average temperature, altitude, and relative humidity were considered as the factor in secondary division.

## Natural succession index

The Natural Succession Index(NSI) designed in our study, derived from the habitat quality section mentioned in circuit theory [28]. The theory focus on the ecological processes in

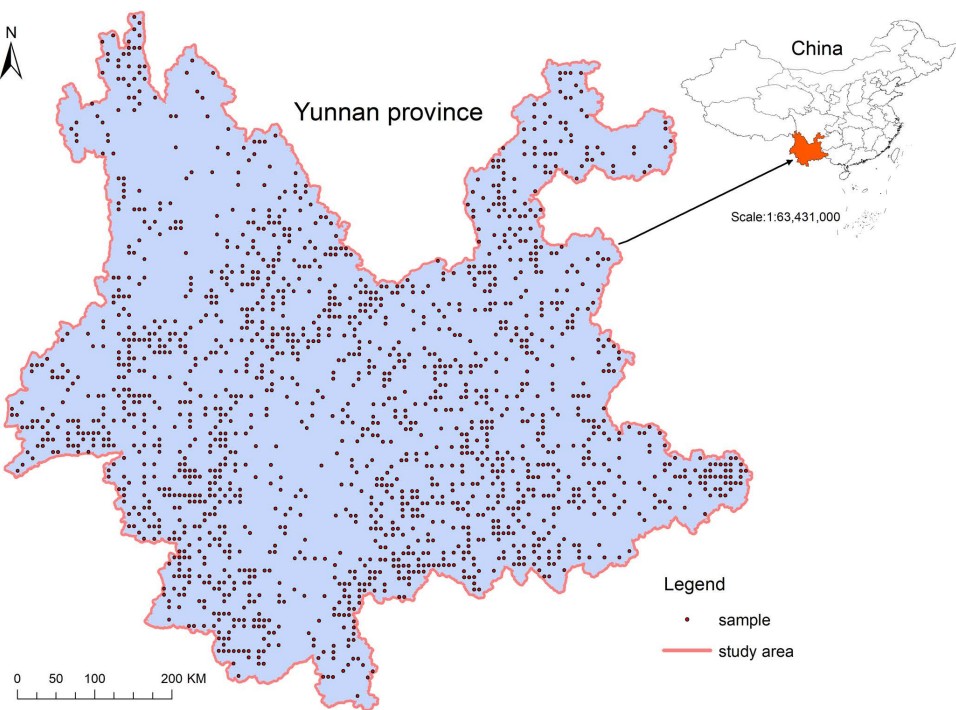

**Fig 1. Map of the study area, Yunnan province, southwest China.** The study area is enclosed by a pink line. Points in the map indicate grassland monitoring sample distribution in 1987. Top right: study area in China. The cases of distribution plotted on the map were derived from the crude data at the province level.

landscape study, and has been widely used in ecological protection analysis [29–31]. The higher habitat quality the environment possess the higher speed the ecological processes has [29,32]. We used the habitat quality mentioned in circuit theory because they all share the ecological processes.

The Natural Succession Index for a specific climatic sub-area is calculated based on the number of points in each succession type as of 2017. This index quantifies the external environmental influence on natural succession. Different sub-areas will exhibit varying results due to distinct climate conditions. However, a general trend is observed: the greater number of points in advanced succession stages, the more favorable climate conditions for natural succession(Fig 2(a)).

Moreover, each stage of succession should be considered because the process of natural restoration is dynamic and involves various stages (secondary bare land stage → grassland stage → shrub stage → forest stage), with each stage playing a distinct role in the restoration process. Some scholars [32,33] regard habitat quality as an indicator of the environment's habitability for organisms. They suggest that areas with higher habitat quality are more productive and stable, making succession easier and more likely to progress to the next stage (Fig 2(b)).

To incorporate this into our analysis, we introduced habitat quality (Q) into our formula to weight each succession stage in the course of natural restoration. Habitat quality (Q) was calculated based on the resistance of the landscape matrix for the functional group of focal species, with Q being the reciprocal of resistance. The resistance values were obtained through bibliographical review [33]. The resistance values of grassland, shrub and forests in Gurrutxaga's(2011) study were 1, 5 and 30. The resistance values of each ecosystem were verified by many studies [34–39] and consultation with five experts in this area by Gurrutxaga's group.

We denote P as the Natural Succession Index(NSI) for different climate condition sub-areas and use the following formula to calculate the value of P for each sub-area. The formula for the natural succession index is based on the form of a power exponent. If P equals 1, it indicates that the area has the maximum natural succession index, suggesting that succession will proceed at an optimal rate (Formula 1).

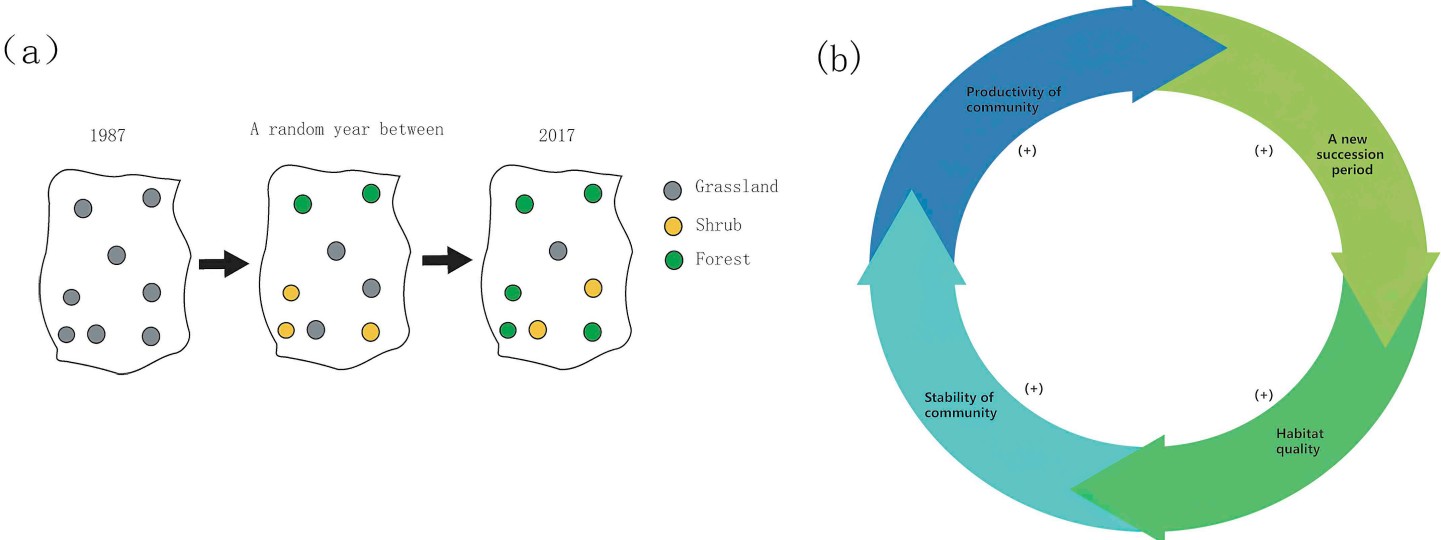

**Fig 2. The ecological processes in study.** (a) Demonstration of succession in one area. (b) Relationship between succession and habitat quality. (+) represents that there is a positive correlation between them.

$$P = \left(Q_1 \times n_1 + \cdots + Q_x \times n_x\right)/N \tag{1}$$

N represents the total number of samples in one climate sub-area, nx represents the total number of samples of each seral stage in the same climate sub-area, Qx represents the habitat quality values of each seral stage.

In our study, we classified our monitoring samples into 3 succession types: 1) Non-succession sample (grassland stage), this land type is always grassland and had never changed from 1987 to 2017. 2) Hard succession sample (shrub stage), this land type had changed, but still kept in shrub stage at 2017. 3) Accessible succession sample (forest stage), this land type had changed, and entered into forest stage at 2017. The numbers of grassland stage, shrub stage and forest stage were used to calculate P value.

The change trend of the Natural Succession Index and the time required to transition from the grassland stage to the forest stage were used to reveal the correlation of Natural Succession Index and time reach to forest stage. The mean times reach to the forest stage were calculated to present the time reach to forest stage in different climate sub-areas. Logarithmic transformations were applied to eliminate the differences of order of magnitude be tween Natural Succession Index and time reach to forest stage for more intuitive exhibition of correlation.

R was used to conduct the analysis. GGplot2 and GGridges packages were used to perform the ranges of environment factors of different climate sub-areas.

## Community characteristics in vegetation succession

A total of 837 samples were selected from the entire samples(1331) to analyze the characteristics of succession. Every sample had reached the forest stage as of 2017. These samples were classified into different climate sub-areas based on the results of the climate sub-area division. Changes in dominant species, stand volume, and succession time in each stage were recorded to characterize the succession in each climate sub-area. The community characteristics were derived from our long-term forest resource monitoring system.

R was used to conduct the analysis. GGplot2 package was used to perform the ranges of environment factors of different climate sub-areas.

## Results

### Climate sub-areas

The study area was initially divided into four climate regions, designated as Region C, Region M, Region W, and Region H (details in Fig 3). Region C encompasses 64 towns located on the southeastern margin of the Tibetan Plateau and is characterized by a cold temperate climate. Region M includes 153 towns situated in the transitional zone between the southeastern margin of the Tibetan Plateau and the Yunnan Plateau, featuring a cool temperate climate. Region W comprises 863 towns in the mid and eastern areas of the Yunnan Plateau, with a warm climate. Region H consists of 338 towns predominantly located south of the Tropic of Cancer, characterized by a tropical climate (details in Fig 3 (a)).

Following the initial division, we conducted a secondary subdivision using the same boundaries and methods. This resulted in the division of the four primary regions into 13 distinct zones, representing the new configuration of climate condition sub-areas in Yunnan (details in Fig 3(b)).

The climate conditions in each sub-area show clear distinctions. The pattern of annual accumulated temperature among the primary divisions follows the order: H > W > M > C. In region C, the annual precipitation displays a notable threshold between sub-areas C-I and C-II, set at 900mm. The maximum relative humidity differs, being 77% in C-I and 85% in

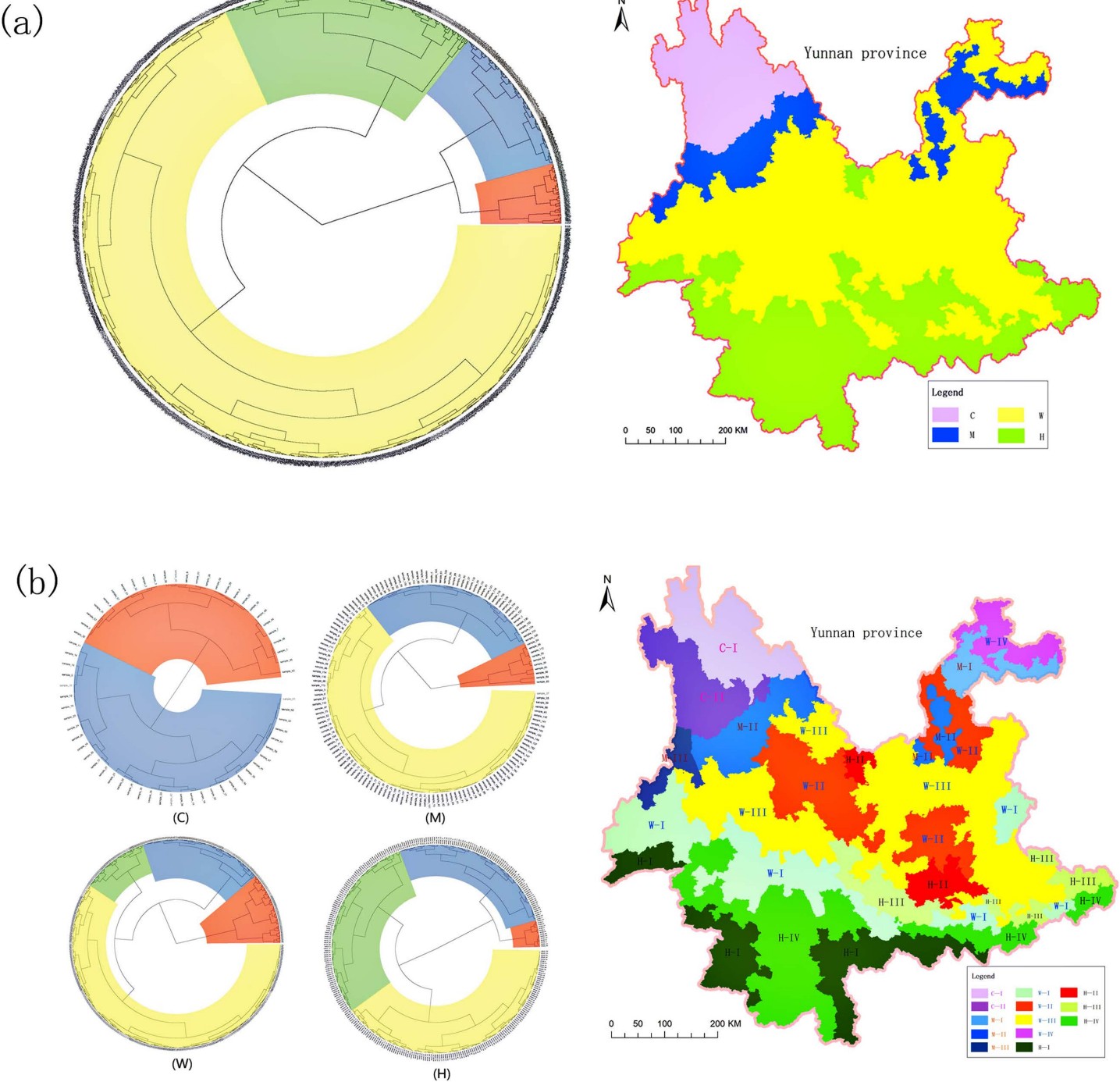

**Fig 3.** the division of climate. (a) Primary division of climate. Left figure is the result of first cluster analysis by annual accumulated temperature. Right figure is the result displays in Yunnan province. (b) Secondary division of climate. Left figure is the results of secondary cluster analysis by rest 4 factors in each. Right figure is the result displays in Yunnan province. The cases of division plotted on the map were derived from the crude data at the town level.

C-II. The minimum mean annual temperature in both sub-areas is 1°C. Differences in mean altitude between C-I and C-II are not significant. In region M, annual precipitation follows the pattern: M-I > M-II > M-III. The relative humidity in M-II is significantly lower than in the

other sub-areas. The minimum mean annual temperature also varies, following the order: M-I > M-II > M-III. The mean altitude in M-I is lower than in the other sub-areas. In region W, the annual precipitation in W-I is significantly higher than in the other sub-areas and exhibits a wide range. The range of relative humidity is lower in W-II compared to the other sub-areas. Differences in mean annual temperature are not significant, but the mean altitude in W-IV is notably lower than in the other sub-areas. In region H, annual precipitation in H-I is considerably higher than in the other sub-areas, with a large range. Both annual precipitation and relative humidity are lower in H-II compared to the other sub-areas. The maximum mean annual temperature in H-II is also lower. The mean altitude in H-II is notably higher than in the other sub-areas. Details are provided in Table 1 and Fig 4.

## Natural succession index and community compositions

We gained the resistance value of different ecosystem types through bibliographical review, which applies to the areas in temperate zone, subtropical zone, or tropical zone. Yunnan province is a region possesses the three climate types. The habitat of three ecosystem types (forest, shrub and grassland) were calculated by resistance valve (Details in Table 2).

After conducting an analysis of Natural Succession Index with 1311 samples in different climate condition sub-areas, we found that the rule of Natural Succession Index of different sub-areas in primary division follows as H > W > M > C, and the rule of Natural Succession Index of different sub-areas in secondary division follows as H-I > H-IV > W-I > M-III > H-III > M-II > W-III > W-IV > C-II > W-II > M-I > H-II > C-I. The distributions of each types details in Table 3.

In region C, the length of grassland stage in region C-I is about double of that in region C-II, and forest volume is lower than region C-II during a relatively similar time. The Stand volumes are between 5m³ to 30m³in region C-I, and 40m³ in region C-II. The key species is Pinus yunnanensis in the two region.

In region M, time of finishing the first two stages is analogous in region M, nevertheless, grassland stage and shrub stage in each climate condition sub-area are different. The Stand volumes are between 20m³ to 50m³in region M-I, between 20m³ to 60m³in region M-II, and 20m³in region M-III. The communities are dominated by pioneer species in region M-I and

**Table 1. Spatial variations.**

| Primary division | Annual accumulated temperature(C°) | Second division | Annual precipitation(mm) | Relevant humidity(%) | Mean annual temperature(C°) | Mean altitude(m) |
|---|---|---|---|---|---|---|
| C | 1489 ~ 3215 | I | 636 ~ 898 | 65 ~ 77 | 1 ~ 10 | 2614 ~ 4069 |
| | | II | 900 ~ 1200 | 65 ~ 85 | 6 ~ 10 | 2582 ~ 3328 |
| M | 3280 ~ 4344 | I | 690 ~ 932 | 74 ~ 84 | 6 ~ 13 | 1452 ~ 2978 |
| | | II | 874 ~ 1135 | 65 ~ 76 | 8 ~ 13 | 2243 ~ 3021 |
| | | III | 1152 ~ 1585 | 73 ~ 81 | 10 ~ 13 | 2066 ~ 2634 |
| W | 4348 ~ 6281 | I | 1166 ~ 1594 | 72 ~ 85 | 13 ~ 19 | 1181 ~ 2224 |
| | | II | 713 ~ 1015 | 63 ~ 76 | 11 ~ 18 | 1305 ~ 2383 |
| | | III | 920 ~ 1240 | 63 ~ 82 | 11 ~ 19 | 1012 ~ 2645 |
| | | IV | 796 ~ 1037 | 75 ~ 85 | 12 ~ 16 | 613 ~ 1646 |
| H | 6302 ~ 8287 | I | 1504 ~ 2253 | 77 ~ 87 | 16 ~ 23 | 334 ~ 1770 |
| | | II | 716 ~ 983 | 63 ~ 75 | 16 ~ 19 | 1231 ~ 1740 |
| | | III | 893 ~ 1266 | 72 ~ 83 | 15 ~ 22 | 450 ~ 1903 |
| | | IV | 1090 ~ 1708 | 71 ~ 86 | 16 ~ 22 | 579 ~ 1634 |

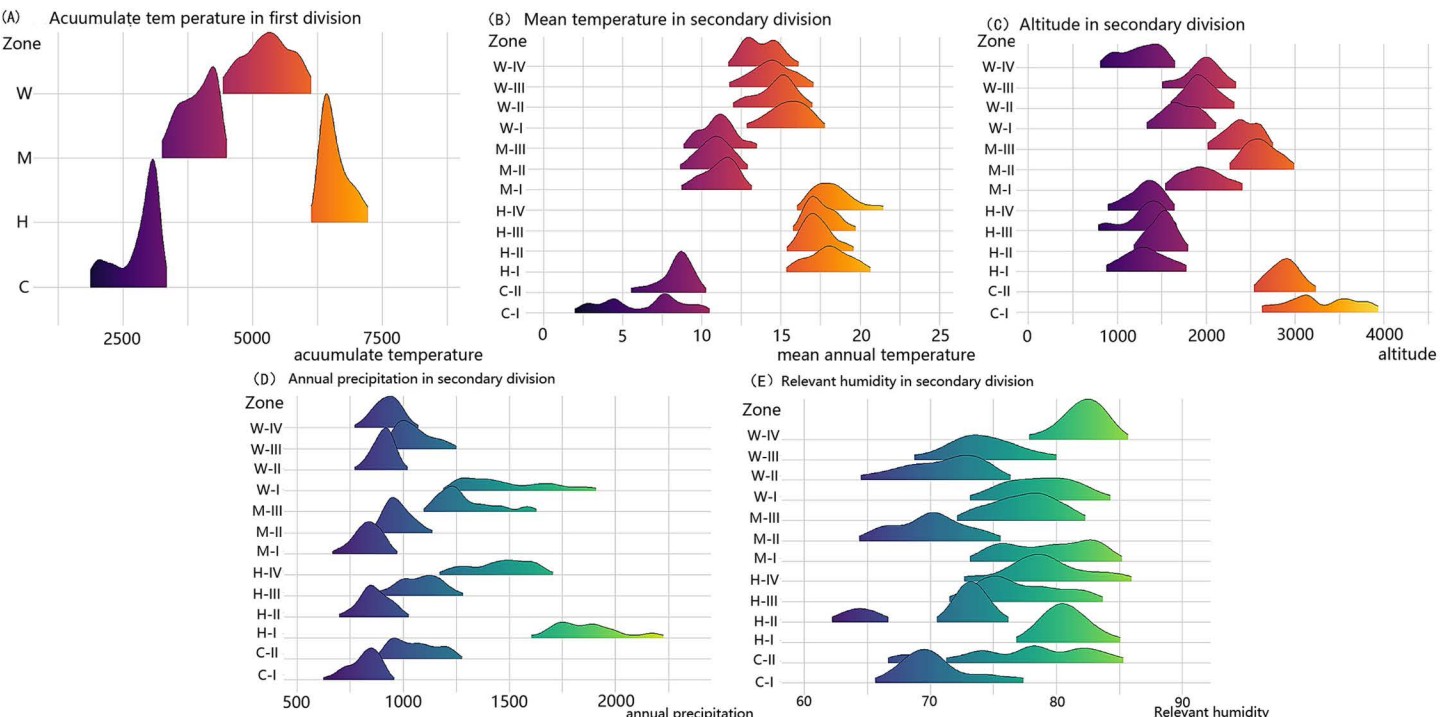

**Fig 4. Spatial variations of each habitat factors.** (A) Spatial variation of accumulate temperature. (B) Spatial variation of mean temperature. (C) Spatial variation of attitude. (D) Spatial variation of annual precipitation. (E) Spatial variation of relevant humidity.

**Table 2. Habitat quality values of each ecosystem type.**

| Ecosystem type | Resistance | Habitat quality |
|---|---|---|
| Forest | 1 | 1 |
| Shrub | 5 | 0.2 |
| Grassland | 30 | 0.0333 |

M-II, such as Pinus yunnanensis, Alnus nepalensis, and so on. The communities are dominated by constructive species in region M-III, Castanopsis delavayi.

In region W, the time of finishing the first two stages in region W-I is same to W-II. The time in region W-III is round 6 years. The time in region W-IV is round 20 years. The Stand volumes are between 40m³ to 70m³ in regionW-I, between 20m³ to 50m³ in region W-II, between 40m³ to 70m³ in region W-III, and between 20m³ to 20m³ in region W-IV. The communities are dominated by pioneer species in region W-II and W-IV, such as Pinus yunnanensis, Alnus nepalensis, and so on. The communities are dominated by constructive species in region W-I and W-III, such as Cyclobalanopsis spp.,Lithocarpus spp., and so on.

In region H, the time of finishing the first two stages in region H-I and H-IV is very short, the results are contrary to the time in region H-II and H-III. The Stand volumes are between 70m³ to 110m³ in region H-I, round 5m³ in region H-II, between 20m³ to 50m³ in region H-III, and between 50m³ to 110m³ in region H-IV. The communities possess a high diversity of trees in region H-I and H-IV. The communities are dominated by saplings with low stability in region H-II. The communities are usually composed of some typical species of deciduous monsoon forest in region H-III. (Details in Table 4, Fig 5).

Table 3. Natural succession index of each sub-area.

| Primary/Secondary division | Congregate | | Grassland stage | | Shrub stage | | Forest stage | | P (Natural Succession Index) |
|---|---|---|---|---|---|---|---|---|---|
| | Numbers | Percent | Numbers | Percent | Numbers | Percent | Numbers | Percent | |
| C | 71 | 100.00 | 21 | 29.58 | 29 | 40.84 | 21 | 29.58 | 0.3873 |
| M | 95 | 100.00 | 32 | 33.68 | 19 | 20.00 | 44 | 46.32 | 0.5144 |
| W | 657 | 100.00 | 120 | 18.27 | 138 | 21.00 | 399 | 60.73 | 0.6554 |
| H | 488 | 100.00 | 75 | 15.37 | 40 | 8.20 | 373 | 76.43 | 0.7859 |
| C-I | 35 | 100.00 | 9 | 25.72 | 20 | 57.14 | 6 | 17.14 | 0.2943 |
| C-II | 36 | 100.00 | 12 | 33.33 | 9 | 25.00 | 15 | 41.67 | 0.4778 |
| M-I | 31 | 100.00 | 13 | 41.94 | 9 | 29.03 | 9 | 29.03 | 0.3624 |
| M-II | 49 | 100.00 | 15 | 30.61 | 8 | 16.33 | 26 | 53.06 | 0.5735 |
| M-III | 15 | 100.00 | 4 | 26.67 | 2 | 13.33 | 9 | 60.00 | 0.6356 |
| W-I | 166 | 100.00 | 32 | 19.28 | 34 | 20.48 | 100 | 60.24 | 0.6498 |
| W-II | 202 | 100.00 | 48 | 23.76 | 94 | 46.54 | 60 | 29.70 | 0.3980 |
| W-III | 268 | 100.00 | 62 | 23.13 | 71 | 26.49 | 135 | 50.38 | 0.5644 |
| W-IV | 21 | 100.00 | 3 | 14.29 | 8 | 38.10 | 10 | 47.61 | 0.5571 |
| H-I | 188 | 100.00 | 26 | 13.83 | 20 | 10.64 | 142 | 75.53 | 0.7812 |
| H-II | 48 | 100.00 | 18 | 37.50 | 18 | 37.50 | 12 | 25.00 | 0.3375 |
| H-III | 95 | 100.00 | 18 | 18.95 | 22 | 23.16 | 55 | 57.89 | 0.6316 |
| H-IV | 157 | 100.00 | 21 | 13.38 | 19 | 12.10 | 117 | 74.52 | 0.7739 |

The time trend of different climate condition sub-areas proceed in forest stage are similar as the Natural Succession Index. The highest Natural Succession Index emerged in region H-I, and the time proceed in forest stage also hit the peak. The lowest Natural Succession Index emerged in region H-II, and the time proceed in forest stage also reach the bottom (Details in Fig 6).

## Discussion

### Climate zone in Yunnan province

In this study, Yunnan Province has been divided into four regions after the first division. Duan et al. [40] utilized a fine-grid method to classify the climate of Yunnan into six climate zones: the north tropical zone, the south subtropical zone, the middle subtropical zone, the north subtropical zone, the temperate zone, and the plateau climate zone. However, since the south subtropical, middle subtropical, and north subtropical zones all fall under the subtropical climate category, there are effectively four distinct climate types. This classification aligns with the first division in our study.

Yunnan Province has been divided into 13 sub-areas following the secondary division. The C region, located on the southeastern margin of the Tibetan Plateau, was characterized by high average altitudes, low annual temperatures, and sporadic permanent glaciers distributed. It has been divided into two sub-areas which located at the two sides(east and west) of Baima Snow Mountains (BSM) separately. Zhang et al. [41] found that the annual mean precipitation on the eastern aspect of the BSM(285.6 mm) is less than that on the western aspect(425 mm). These findings support the climate characteristics of sub-regions C-I and C-II, which are cold-dry climate(C-I) and cold-moist climate(C-II), respectively.

The M region forms a belt shape, dispersedly located in the transition zone between the southeastern margin of the Tibetan Plateau and the Yunnan Plateau, as well as regions of high altitude and latitude within the Yunnan Plateau. It has been divided into three sub-areas, influenced by the segmentation of high mountains such as the Gaoligong Mountains,

**Table 4. Seral stage of each climate sub-area.**

| Secondary division | Grassland stage | Shrub stage | Forest stage | Stand volume | Community characteristics |
|---|---|---|---|---|---|
| C-I | 10 years to 13 years | 5 years to 6 years | 11 years to 15 years | 5m³ to 30m³ | The vegetation in this region includes *Pinus yunnanensis* forests and *Pinus yunnanensis* + *Quercus pannosa* forests, both of which are primarily dominated by *Pinus yunnanensis* as a pioneer species. |
| C-II | 5 years | 9 years | 16 years | 40m³ | There is *Pinus yunnanensis* + *Quercus acutissima* forest which are dominated by *Pinus yunnanensis* as a pioneer species. |
| M-I | round 4 years | round 12 years | round 16 years | 20m³ to 50m³ | The communities in this region are dominated by pioneer species such as *Pinus yunnanensis* and various plants from the Lauraceae, Juglandaceae, and Anacardiaceae families, including *Litsea pungens, Rhus chinensis, Litsea cubeba,* and *Juglans cathayensis* etc. |
| M-II | round 9 years | less 5 years | round 16 years | 20m³ to 60m³ | The communities are typically dominated by pioneer species such as *Pinus yunnanensis* and *Alnus nepalensis*, with a smaller presence of constructive species like *Quercus pannosa, Quercus semicarpifolia*, and *Picea likiangensis*. Only a few communities are predominantly composed of these constructive species. |
| M-III | 8 years | 8 years | 14 years | 20m³ | There are *Castanopsis delavayi* + *Alnus nepalensis* forest which are dominated by *Castanopsis delavayi* saplings as constructive species. |
| W-I | round 5 years | less 7 years | round 18 years | 40m³ to 70m³ | Communities are typically dominated by constructive species such as *Cyclobalanopsis* spp., *Lithocarpus* spp., and *Quercus* spp., along with companion species from the *Lauraceae* and *Theaceae* families. Only a few communities are primarily dominated by species from *Lauraceae* and *Theaceae*, including *Schima wallichii, Phoebe zhennan*, and *Eurya groffii*. |
| W-II | round 9 years | less 3 years | round 18 years | 20m³ to 50m³ | Communities are generally dominated by pioneer species such as *Pinus yunnanensis* and *Alnus nepalensis*, alongside some constructive species including *Cyclobalanopsis* spp., *Lithocarpus* spp., and *Keteleeria evelyniana*. Only a few communities are predominantly dominated by *Cyclobalanopsis* spp. and *Lithocarpus* spp.. |
| W-III | less 3 years | round 3 years | round 25 years | 40m³ to 70m³ | Communities are typically dominated by constructive species such as *Cyclobalanopsis* spp. and *Quercus* spp., with companion species including *Pinus yunnanensis, Schima argentea*, and various species of *Lauraceae*. Only a few communities are predominantly dominated by pioneer species such as *Pinus yunnanensis* and *Alnus nepalensis*. |
| W-IV | round 10 years | 8 years to 12 years | round 10 years | 20m³ to 30m³ | Communities are typically dominated by pioneer species such as *Alnus nepalensis, Rhus chinensis, Litsea mollis, Paulownia fortunei*, and *Cerasus yunnanensis*, along with constructive species from the *Cyclobalanopsis* spp. and the *Lauraceae* family. |
| H-I | round 3 years | less 3 years | round 25 years | 70m³ to 110m³ | Communities here possess a high diversity of trees consisting of 26 families(*Theaceae,Euphorbiaceae,Betulaceae,Leguminosae,Aquifoliaceae,-Juglandaceae,Lauraceae,Actinidiaceae,Flacourtiaceae,Fagaceae, Moraceae,Myrtaceae,Magnoliaceae*, and so on.),furthermore,plants in *Fagaceae* are mainly *Cyclobalanopsis* spp. and *Castanopsis* spp. |
| H-II | 10 years to 15 years | 5 years to 15 years | 5 years to 10 years | round 5m³ | Communities here are co-dominated by *Quercus acutissima* saplings, *Pinus yunnanensis*, and *Broussonetia papyrifera*. |
| H-III | round 11 years | less 5 years | round 14 years | 20m³ to 50m³ | Communities are usually composed of some typical species of deciduous monsoon forest such as *Fagaceae* (*Lithocarpus* spp. is far superior to others), *Pinus yunnanensis, Keteleeria evelyniana, Ficus cyrtophylla, Eriolaena spectabilis, Bauhinia acuminata, Albizia kalkora, Eurya groffii, Phyllanthus emblica,* and so on. |
| H-IV | round 3 years | less 2 years | round 14 years | 50m³ to 110m³ | Communities here possess a high diversity of trees, and the species are similar to H-I region, but plants in *Fagaceae* are rarely seen except *Cyclobalanopsis* spp.. |

Dashanbao Mountains, and Yaoshan Mountains [42–44]. The distribution of the M region overlaps significantly with the temperate zone in Yunnan [45], resulting in cool weather conditions. Regions M-I and M-II were similar as the climate characteristics of Yunnan

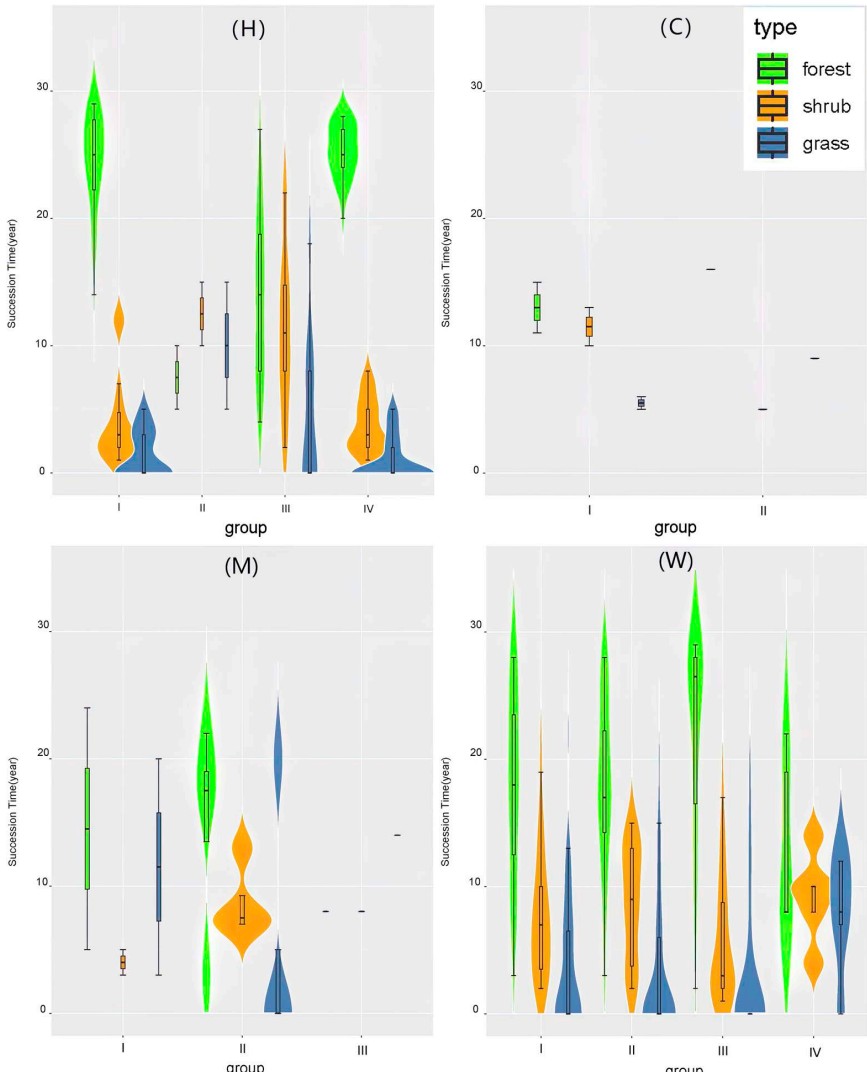

**Fig 5.** Seral stage of each climate sub-areas. C. Contributions of region C-I and region C-II. M. Contributions of region M-I, region M-II and region M-III. W. Contributions of region W-I, region W-II, region W-III and region W-IV. H. Contributions of region H-I, region H-II, region H-III and region H-IV. Green means how long the forest stage keeps in one monitoring point. Yellow means how long the shrub stage would spend to be forest in one monitoring point. Blue means how long the grass stage would spend to be shrub in one monitoring point.

Plateau(dry and cool) [46], but they are colder than Yunnan Plateau climate due to their higher latitude and altitude locations [46]. Region M-III was situated at the west of Gaoligong Mountains, influenced by the warm and wet currents from the Indian Ocean, resulting in a cool and moist climate [47].

The W region is situated in the Yunnan Plateau, characterized by the typical Yunnan Plateau climate, with average temperatures ranging from 12 to 18°C, annual precipitation between 1000 mm and 1200 mm, dry air, long hours of sunlight, a wet season from May to October, and a dry season from November to April [46]. Due to varying landforms and distributions, the area has been divided into four sub-areas, and each of them exhibit distinct characteristics. Region W-I, adjacent to the southwest hilly area of Yunnan, is influenced by the climbing air currents from the Indian Ocean, resulting in relatively high precipitation [48].

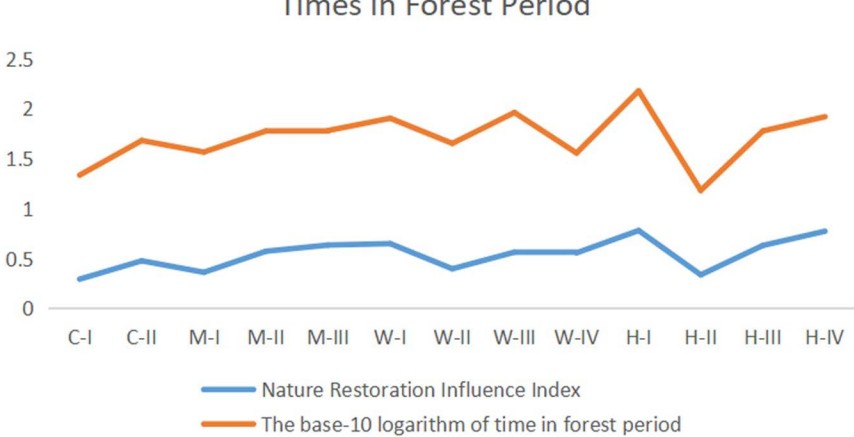

**Fig 6. The trends of Natural Succession Index and Times reach to forest stage.** The blue line is the trends of Natural Succession Index. The red line is the base-10 logarithm of time in forest stage.

Region W-II primarily encompasses basins, dry-warm valleys, and dry-hot valleys, characterized by low precipitation and high evaporation [49]. The climate features of region W-III closely resemble the typical Yunnan Plateau climate. Region W-IV is located in the transition zone of the Yunnan-Guizhou Plateau, with high air humidity [50].

The H region, located in the southern part of Yunnan, is characterized by low altitude and latitude, providing favorable conditions for restoration. However, a small portion of the region falls within dry-hot valleys [49] and areas undergoing desertification [51,52]. Region H-I is the primary distribution area of tropical rainforests, characterized by heat, humidity, and high precipitation [40]. Region H-II consists mostly of dry-hot valleys and desertified land [49,51], suffering from severe dry-hot stress due to a lack of rainfall. Region H-III is situated in the Yuanjiang basin, also a dry-hot valley climate, but with relatively higher precipitation compared to H-II. Region H-IV is the main distribution area of humid evergreen broad-leaved forests and tropical rainforests [53]. While its features are similar to H-I, it receives less precipitation due to its lower altitude.

## Natural restoration in climate sub-areas

In this study, vegetation succession was utilized to quantify the effectiveness of natural restoration under different climate conditions. Cindy et al. [54] also highlight succession dynamics as a key factor in evaluating secondary succession in natural subtropical montane evergreen broad-leaved forests. However, unlike their focus on changes in pioneer species, our study emphasizes succession time as a critical metric.

Tong et al. [55] employed a time series trend analysis to examine long-term vegetation dynamics in the karst areas of southwest China, thereby quantifying the effectiveness of ecological restoration projects in this region. The study indicated a low Project Effectiveness Index (PEI) value [55] for Yunnan province, suggesting that ecological restoration projects there have had limited success. These karst areas predominantly fall within regions C, M, W-IV, H-II, and H-III in our study, all of which exhibit a low Natural Succession Index (NSI). The Project Effectiveness Index (PEI) was developed for quantifying the effectiveness of ecological restoration projects, but it was calculated by the changes of

satellite image, and just evaluated by the land came to green or not, farmland also came green in right season in satellite image. Furthermore, the calculation of index (PEI) was subjected to the success rate of projects, but the success could be influenced by many factors, the environment just one section to decide a successful restoration. In contrast to this, Natural Succession Index (NSI) possesses a wild application prospects, because the index calculated the influence from environment for vegetation growing. Very project involves in vegetation growing could consider it(NSI) to avoid the waste comes from climate impact.

Andrew Willson [56] conducted a study in Xiaozhongdian, located in the northwest of Yunnan, using long-term LULC data. The study revealed that 18% of grassland changed into shrubland from 1981 to 1990, and 53% from 1990 to 1999. Xiaozhongdian falls within the C-II region in our study, and the duration of the shrub stage aligns with our findings. Furthermore, significant correlations between vegetation growth and precipitation were detected in region C [57], supporting our results. Specifically, the grassland stage in C-I is approximately twice as long as in C-II, and forest volume in C-I is lower than in C-II.

The duration of the first two succession stages is analogous across region M, however, the lengths of the grassland and shrub stages vary among the different climate condition sub-areas. This phenomenon can be attributed to differences in geographical units and floristic compositions. Region M-I is situated on the Yunnan Plateau at a relatively lower altitude and belongs to the Sino-Japanese forest flora. In contrast, regions M-II and M-III are located in the transitional zone between the southeastern edge of the Tibetan Plateau and the Yunnan Plateau, and are part of the Sino-Himalayan flora of China [58].

The climate conditions of the sub-areas in region W generally align with those of the Yunnan Plateau. The time required to complete the grassland and shrub succession stages in region W is shorter than in regions C and M. This finding is consistent with a study conducted from 1998 to 2019, which revealed a significant increase in vegetation in the northeast (Zhaotong) and central (Kunming) areas of Yunnan, while vegetation in the Northwest Transverse Range region (regions C and M) showed an insignificant increase [59]. However, exceptions exist due to varying local conditions.

Region W-II is predominantly characterized by dry-warm and dry-hot valleys in the Jingshajiang River basin, areas with high evapotranspiration rates that contribute to a prolonged grassland succession stage [60,61]. Despite this, it takes only about three years to transition to the forest stage once succession reaches the shrub stage. Additionally, region W-IV belongs to the Karst Desertification Region of Yunnan Province, which suffers from severe soil erosion, further extending the grassland stages [51]. Fast-growing and dense-closing shrub species, such as *Rosa omeiensis* Rolfe and *Rubus inopertus* (Diels) Focke, inhibit the growth of tree species for an extended stage once the shrub stage is established, contributing to the prolonged duration of the shrub stage [62].

Region H is characterized by low altitude and high average annual temperatures, with most areas enjoying a warm and humid climate [53]. This climate facilitates high succession speeds and low succession difficulty. However, some challenges persist due to relatively harsh standing conditions. Region H-II and region H-III are part of the karst rocky desertification area and dry-hot valleys in Mengzi, Yuanmou, and Yuanjiang [60,63,64]. The soil in these regions lacks sufficient water retention capacity, resulting in a prolonged grassland stage of 10 to 15 years. The higher precipitation in region H-III compared to region H-II leads to different durations of the shrub stage between these two sub-areas.

## Implication for ecological restoration management

The report by Credit Suisse and McKinsey [65] highlighted that maintaining healthy terrestrial and marine ecosystems requires an annual investment of approximately $300-400 billion worldwide. However, only about $52 billion is actually allocated to nature. Additionally, investment in nature comprises a mere 0.02% of global investment annually. Despite this modest allocation, the returns generated can be approximately 2,500 times the initial investment. Clearly, increasing investment in nature has the potential to yield immeasurable returns.

China has implemented numerous policies for ecological restoration and protection, including the Grain-for-Green Program, the Natural Forest Protection Program, and the Ecological Conservation and Restoration of Life Community [66]. In 2016, China issued a policy prohibiting the logging of natural forests and provided an annual subsidy of $37.2 per hectare as compensation to stakeholders. Additionally, the policy created management and protection jobs for those willing to participate in forest protection within the restricted areas. The Chinese government places significant emphasis on investment in ecological restoration, with government funding increasing annually [65,67]. However, despite these efforts, numerous environmental groups and individuals [68,69] continue to advocate for greater investment in ecosystem protection and restoration from both governmental and non-governmental organizations due to the substantial investment gap needed to restore our environment.

The Natural Succession Index (NSI) offers a method to mitigate the disparity between the funds invested in ecological restoration and the actual financial needs of the environment. By utilizing the NSI of each sub-area in Yunnan Province as a parameter, we can re-balance ecological restoration funds without increasing the overall investment, thereby maximizing ecological returns, as outlined in Formula 2. To achieve an accurate allocation of restoration investment with a fixed input, we calculated the mean NSI value across all climate condition sub-areas, denoted as K, serving as the threshold for fund re-balancing. Sub-areas with an NSI greater than K are considered investment surplus areas, and their funding should be reduced. Conversely, sub-areas with an NSI less than K are considered investment-deficient areas, necessitating compensation from the surplus. The re-balancing of funds can be achieved by following the specified formula (Details in Fig 7).

$$I = I_n \times \left(1 + \frac{K - Px}{K}\right) \qquad (2)$$

In this formula, we take I as a newly indeterminate investment amount, In as the investment amount under different policies, Px as the Natural Succession Index of different climate condition sub-areas, K as the mean value of Natural Succession Index in all climate condition sub-areas.

## Uncertainties, limitations and future perspectives

We acknowledge that this study has certain limitations and uncertainties. First, our analysis was confined to the stage from 1987 to 2017. This raises concerns about the validity of our findings given the limited monitoring stage of 30 years. Additionally, questions may arise regarding why we chose the 30th year as the point for calculating the Natural Succession Index (NSI), rather than other potential benchmarks such as the 20th, 60th, or 150th year. However, Stuart and Carld [70] reported that biomass and species diversity trends stabilize around the 30th year in secondary succession. Similarly, Dieter et al. [71] observed in their study, which included nine groups at various stages of succession (4, 15, 20, 25, 36, 50, 60, 80, 120 years), a critical turning point between the 25th and 36th years where species richness increased at the

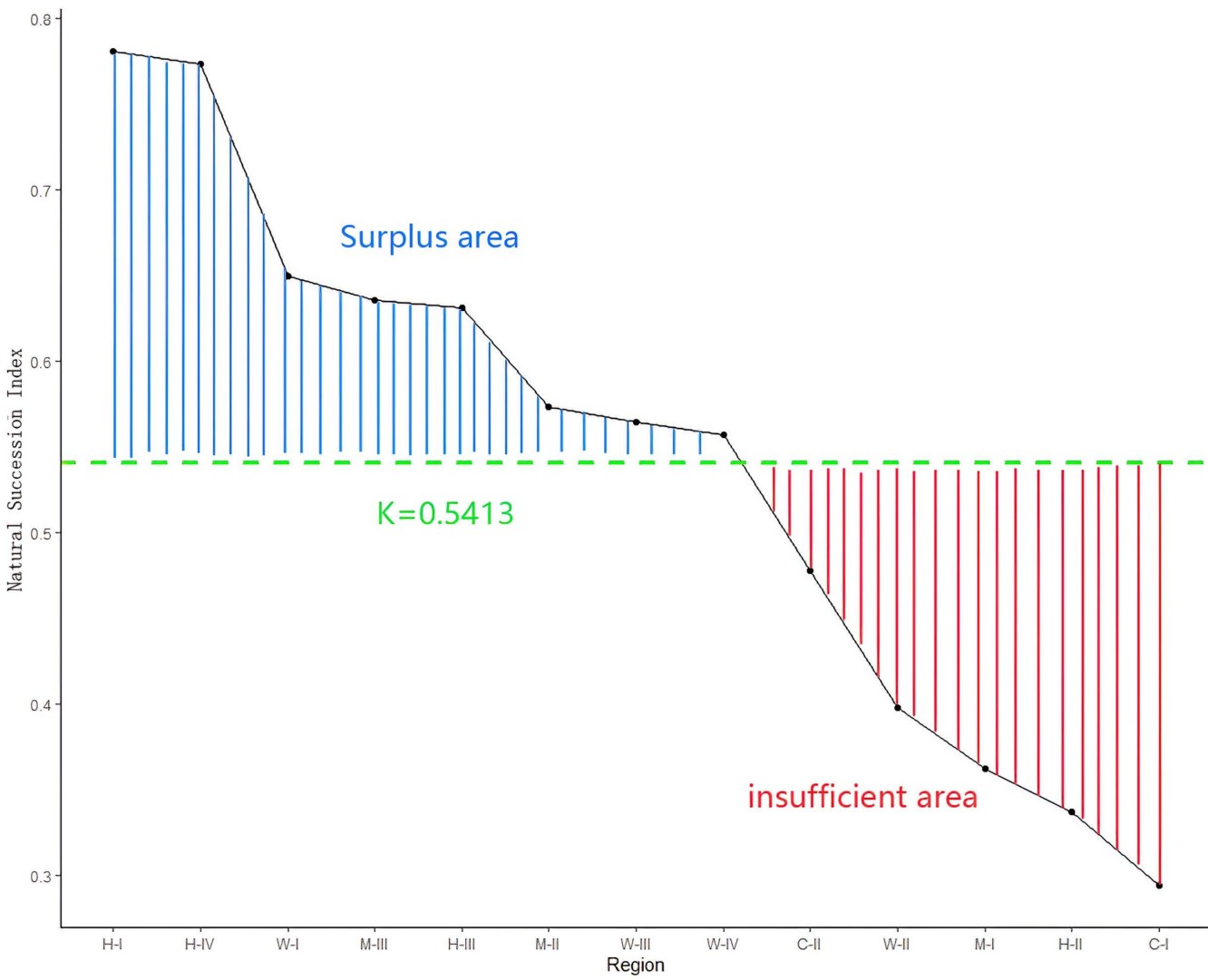

**Fig 7. Supplementary distribution of each climate condition sub-area.**

highest rate and then stabilized. Zhao et al. [72] also found that total soil nitrogen and plant diversity reached a steady state around the 30th year in secondary succession. These findings suggest that the 30th year is a pivotal point where secondary succession enters a stable state, thereby supporting the reliability of our chosen timeframe for calculating the NSI.

Second, our calculation of the NSI involved three seral stages (grassland, shrub, and forest stages). However, our results might lack precision as we only considered the growth form of dominant species (herbaceous, shrub, and woody plants). Other studies have taken more nuanced approaches to classify seral stages, offering a potential avenue for enhancing the accuracy of our calculations. For instance, Bornkamm [73] divided secondary succession into five stages based on rates of floristic change; Kalacska [74] used average height, number of layers, and NDVI range to subdivide forest succession; Helle [75] classified secondary succession into five stages by total turnover rate; and Gong Zhi-wen et al. [76] segmented secondary succession into five stages using importance value and interspecific associations. Addressing

this limitation in future work, we aim to develop a more comprehensive equation that incorporates additional seral stages to enhance the precision of our NSI.

## Conclusions

The post-2020 global biodiversity framework call for extensive ecosystem restoration actions. China made many efforts to restore ecosystem in degenerate by natural restoration. However, there is a generally low efficiency in ecosystem restoration actions, partly due to the neglect of environmental influence on natural restoration in different regions. It's natural restoration works everywhere. To overcome such challenges in this study we proposed a index considering the succession process of natural restoration to improve the planning of natural restoration actions in order to both enhance their effectiveness, as well as inform the allocation of scarce human and financial resources.

In our study, Yunnan province has been divided into 14, and Natural Succession Index in each climate sub-areas were calculated. The rule of Natural Succession Index in different sub-areas follows as H-I > H-IV > W-I > M-III > H-III > M-II > W-III > W-IV > C-II > W-II > M-I > H-II > C-I, which shown the order of feasibility for natural restoration because of the positive correlation between them. Based on the understanding of Natural Succession Index, we developed a algorithm to optimize the investment allocation for natural restoration, thus promoting ecological returns. Our study recommended that future the restoration actions could act upon the restoration feasibility for stepwise implementation, and be prioritized in region easy to restore(region H-I, region H-IV, region W-I, region M-III, and region H-III). However, expanding investment in restoration actions for achieving large afforestation area is not recommended until other regions have been restored in region C-I and region H-II. The investment of restoration could be reallocate by the Natural Succession Index, but constant vigilance is required in the first 5 years after restoration actions implemented to avoid failure from calculated error. Future emphasis should be given to the development of the threshold of Natural Succession Index to improve the reliability in restoration actions.

## Supporting information

**S1 Table. Change of covers in monitoring point.**
(CSV)

**S2 Table. Environment factors for division.**
(CSV)

**S3 Appendix. Codes for creating figure.**
(DOC)

## Acknowledgments

We thank Jie Yang, PHD, Ms Hui Xiong for giving precious recommendations in writing the draft, and the experts who participated in the monitoring work in the last 30 years.

## Author contributions

**Conceptualization:** Weifeng Gui, Qingzhong Wen, Wenyuan Dong.

**Data curation:** Weifeng Gui.

**Investigation:** Weifeng Gui, Qingzhong Wen, Xue Ran.

**Methodology:** Weifeng Gui.

**Project administration:** Qingzhong Wen.

**Resources:** Dechang Kong.

**Software:** Xiaosong Yang.

**Supervision:** Wenyuan Dong, Guangqi Zou.

**Writing – original draft:** Weifeng Gui, Qingzhong Wen, Wenyuan Dong, Xue Ran.

**Writing – review & editing:** Weifeng Gui.

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
