## [Decision Letter · Decision Letter 0]

14 Nov 2024

PONE-D-24-44569Using Spontaneous Vegetation Succession to Evaluate How Natural Restoration Works under Different Climate in Yunnan, Southwest ChinaPLOS ONE

Dear Dr. Gui,

Thank you for submitting your manuscript to PLOS ONE. After careful consideration, we feel that it has merit but does not fully meet PLOS ONE’s publication criteria as it currently stands. Therefore, we invite you to submit a revised version of the manuscript that addresses the points raised during the review process.

We look forward to receiving your revised manuscript.

Kind regards,

Ghulam Yaseen, Ph.D.

Academic Editor

PLOS ONE

4. We note that Figures 1 and 3 in your submission contain [map/satellite] images which may be copyrighted. All PLOS content is published under the Creative Commons Attribution License (CC BY 4.0), which means that the manuscript, images, and Supporting Information files will be freely available online, and any third party is permitted to access, download, copy, distribute, and use these materials in any way, even commercially, with proper attribution. For these reasons, we cannot publish previously copyrighted maps or satellite images created using proprietary data, such as Google software (Google Maps, Street View, and Earth). For more information, see our copyright guidelines: http://journals.plos.org/plosone/s/licenses-and-copyright.

1. You may seek permission from the original copyright holder of Figures 1 and 3 to publish the content specifically under the CC BY 4.0 license. 

Reviewers' comments:

Reviewer's Responses to Questions

**Comments to the Author**

1. Is the manuscript technically sound, and do the data support the conclusions?

Reviewer #1: Yes

Reviewer #2: Partly

2. Has the statistical analysis been performed appropriately and rigorously? 

Reviewer #1: Yes

Reviewer #2: I Don't Know

3. Have the authors made all data underlying the findings in their manuscript fully available?

Reviewer #1: No

Reviewer #2: No

4. Is the manuscript presented in an intelligible fashion and written in standard English?

Reviewer #1: No

Reviewer #2: No

5. Review Comments to the Author

Reviewer #1: Using Spontaneous Vegetation Succession to Evaluate How Natural Restoration Works under Different Climate in Yunnan, Southwest China

The present work deals with the use of Spontaneous Vegetation Succession in China to Evaluate How Natural Restoration Works under Different Climate. First of all, if authors would like to present their study in a more scientific format, some title words should be replace to more appropriate concepts. “Spontaneous vegetation” could be replaced to “herbaceous vegetation” or “successional vegetation” since the word “spontaneous” is not a functional or structural concept of vegetation. In a similar way, the word “works” is not a proper word to refer to ecosystem restoration activity. Perhaps the word “proceeds” or “progress” should adheres better.

The short title does not read short.

Keywords:

Natural restoration; Spontaneous vegetation succession: It is strongly recommended that there should not be the same title words but different ones contained in the document or refer to the study topic, that helps to retrieve the paper in addition to the title words solely.

Climate condition sub areas; Index quantify; Community characteristic: these keywords are not clear or properly written or the authors have not clarity on the concepts they want to consider.

Abstract

“Yet, there remain uncertainties regarding with natural restoration, for example, how to confirm natural restoration works for different climate conditions, and how could we quantify it.”

Natural restoration has been widely and for long time studied, formerly in temperate climates and there after in tropical and many other climates. Then, this motivation is not sufficiently strong.

However, the aim of quantifying Natural Succession from 1703 succession monitoring data(1987 to 2017) with habitat quality data (14 sub-areas) in different Seral stages(grassland, shrub and forest), is a novelty and interesting research.

The word “spontaneous” simply is herein used as a synonymy of the word “natural”, so it could be better to use the longstanding term of natural used in this literature.

“The rule of it follows as H-I > H-IV > W-I > M-III > H-III > M-II > W-III > W-IV > C-II > WII > M-I > H-II > C-I. The times of spontaneous succession in forest stage from grassland stage in each sub-area are similar as the rule of index. The dominate species in each sub-area are different. The stand volumes are influenced by the index, and showed a positive correlation. Future, the index could be used to reallocate the investment in natural restoration project for better returns. Constant vigilance is required in the first 5 years after restoration actions implemented to avoid failure from calculated error.”

The above paragraph is not clear. It needs to be rewritten. The rule (letters and “numbers” or categories) shows there does not say where or how it comes from to many readers not specialized in this topic.

The abstract does not show clearly the Results or the quantitative data obtained in the study. It only limits to three description lines without figure numbers. Perhaps provide ranges of data from Table 3. Stand volumes of vegetation are key results of regeneration and may be shown here.

After reading methodology and Results, I see a problem in summarizing the key results into the abstract. Authors should attend this observation to give the reader a comprehensible understanding of what these letters and categories mean.

Introduction

Line 41. At the end replace “M” for “m”.

Line 45 – 53. Are not fluently reading.

Materials and methods

Line 89. The word “investigative” could be replaced for “research”.

The number of shared characteristics of the samples are 4 or 5?

Line 116. It should be breifley explained how is annual accumulated temperature obtained though is not a common variable used. Is it a daily summarizing of temperature values (maximum day value or which temperature value) or how are the temperature values summarized throughout the year.

Could be useful to mention which graphical software was used to make figures 3-5.

Results

Line 202. Fig. 4. Should be better to delete “It’s a figure to show the” and simply start the text with “Spatial variation of……”.

Line 210, 211. Text already presented in Table 2, so decide which way to present Resistance and Habitat quality data, in a paragraph or a table. Rewrite the paragraph accordingly to the decision.

Table 3. Use of percent or percentage? Correct.

Table 4. Make shorter community characteristics column and expand the previous four columns for better reading.

Discussion

Line 292. Tang does not match with number [54]. Correct.

Use italics for the scientific names.

df

Reviewer #2: The manuscript PONE-D-24-44569 contains important findings that contribute to our understanding of spontaneous or natural" vegetation succession across different climatic conditions in Yunnan, China. The findings will be of significant interest to the broad scientific community and are backed by a strong data set spanning 30 years of observations.

However, this does not come through from the abstract nor the introduction, and it is hard to understand from the current presentation of the methods.

In order to better understand and appreciate the findings and implications of this study a better organization of the methods is needed, which then can be used to re-write the abstract and discussion. For example, the abstract does not do justice to the span of sites, observations, and intervals of the observation used in this study.

Only after reading halfway into the manuscript is enough information presented on the 30-year data set that has 6 observations taken at five-year intervals.

The authors are suggested to please improve readability of the text narrative by providing information on the study design and how locations-sites/plots/samples/time intervals. As it is currently presented it does not convey the importance and complexity of the results being present, and it even detracts from appreciating the implications from the findings.

Please add to Table 1 a blank row to provide clarity and space - between the "Primary division" categories and for Table 4 adding a blank row to provide clarity and space - between the "Secondary division" categories.

The information of Figure 2 and Figure 3

6. PLOS authors have the option to publish the peer review history of their article (what does this mean? ). If published, this will include your full peer review and any attached files.

**Do you want your identity to be public for this peer review?** For information about this choice, including consent withdrawal, please see our Privacy Policy .

Reviewer #1: **Yes: ** Jose Luis Martinez Sanchez

Reviewer #2: No

---

## [Author Response · Author response to Decision Letter 1]

17 Dec 2024

Response to reviewer 1:

Dear professor，

Thanks for your precious advice. We deeply appreciate your meticulous and highly professional work. These advice possess great significance for improving the quality of our manuscript. We attached great importance to these, and took actions carefully to revise our manuscript. In this response letter we attached the advice for our manuscript and made the response under it. Furthermore, we resubmitted the revised manuscript to submission system, and the texts we had revised were marked in red in the new one. The details of modification below as:

Q1:The present work deals with the use of Spontaneous Vegetation Succession in China to Evaluate How Natural Restoration Works under Different Climate. First of all, if authors would like to present their study in a more scientific format, some title words should be replace to more appropriate concepts. “Spontaneous vegetation” could be replaced to “herbaceous vegetation” or “successional vegetation” since the word “spontaneous” is not a functional or structural concept of vegetation. In a similar way, the word “works” is not a proper word to refer to ecosystem restoration activity. Perhaps the word “proceeds” or “progress” should adheres better.

A:Dear professor, thanks for your advice, and the word “spontaneous vegetation succession” in our draft has been replaced to “natural vegetation succession” which was the longstanding term of natural used in this literature.

Q2:The short title does not read short.

A:Dear professor, thanks for your advice, and we have modified our short titles in short, and kept the main idea we want to convey as possible.

Keywords:

Q3:Natural restoration; Spontaneous vegetation succession: It is strongly recommended that there should not be the same title words but different ones contained in the document or refer to the study topic, that helps to retrieve the paper in addition to the title words solely.

Climate condition sub areas; Index quantify; Community characteristic: these keywords are not clear or properly written or the authors have not clarity on the concepts they want to consider.

A:Dear professor, thanks for your advice, and we have changed the keywords into “Herbaceous vegetation succession; Natural succession index; Community composition; Climate division; Long term monitoring” to exhibit the changes of herbaceous vegetation succession in our long term monitoring in community level.

Abstract

Q4:“Yet, there remain uncertainties regarding with natural restoration, for example, how to confirm natural restoration works for different climate conditions, and how could we quantify it.”

Natural restoration has been widely and for long time studied, formerly in temperate climates and there after in tropical and many other climates. Then, this motivation is not sufficiently strong.

However, the aim of quantifying Natural Succession from 1703 succession monitoring data(1987 to 2017) with habitat quality data (14 sub-areas) in different Seral stages(grassland, shrub and forest), is a novelty and interesting research.

A:Dear professor, thanks for your advice, and we have dropped the description which without sufficient support to improve motivation in this section. Furthermore, we changed the description to a new one contain the main idea of quantifying natural succession from a long-term succession monitoring data habitat quality data in different succession stages.

Q5:The word “spontaneous” simply is herein used as a synonymy of the word “natural”, so it could be better to use the longstanding term of natural used in this literature.

A:Dear professor, thanks for your advice, and we have changed the word “spontaneous” into the word “natural”.

Q6:“The rule of it follows as H-I > H-IV > W-I > M-III > H-III > M-II > W-III > W-IV > C-II > WII > M-I > H-II > C-I. The times of spontaneous succession in forest stage from grassland stage in each sub-area are similar as the rule of index. The dominate species in each sub-area are different. The stand volumes are influenced by the index, and showed a positive correlation. Future, the index could be used to reallocate the investment in natural restoration project for better returns. Constant vigilance is required in the first 5 years after restoration actions implemented to avoid failure from calculated error.”

The above paragraph is not clear. It needs to be rewritten. The rule (letters and “numbers” or categories) shows there does not say where or how it comes from to many readers not specialized in this topic.

The abstract does not show clearly the Results or the quantitative data obtained in the study. It only limits to three description lines without figure numbers. Perhaps provide ranges of data from Table 3. Stand volumes of vegetation are key results of regeneration and may be shown here.

After reading methodology and Results, I see a problem in summarizing the key results into the abstract. Authors should attend this observation to give the reader a comprehensible understanding of what these letters and categories mean.

A:Dear professor, thanks for your advice, and we have rewrote this paragraph in our abstract. First, we added the description for where 14 categories come from. Second, we added the nature succession index value adhered to each sub-area in the order of sub-areas. Finally, we provided ranges of time and stand volumes, and added the relations between them and nature succession index value.

Introduction

Q7:Line 41. At the end replace “M” for “m”.

A:Dear professor, thanks for your advice, and we have replaced “M” for “m”.

Q8:Line 45 – 53. Are not fluently reading.

A:Dear professor, thanks for your advice, and we have rewrote this paragraph for better readability.

Materials and methods

Q9:Line 89. The word “investigative” could be replaced for “research”.

A:Dear professor, thanks for your advice, and we have replaced “investigative” for “research”.

Q10:The number of shared characteristics of the samples are 4 or 5?

A:Dear professor, thanks for your advice, and we have revised the mistake in our draft, the number of shared characteristics among the samples is now 5.

Q11:Line 116. It should be breifley explained how is annual accumulated temperature obtained though is not a common variable used. Is it a daily summarizing of temperature values (maximum day value or which temperature value) or how are the temperature values summarized throughout the year.

A:Dear Professor, thank you for your advice. Accumulated temperature represents the total sum of daily mean temperatures that exceed 0°C, accumulated over a specified temporal interval. In our study, we utilized the total sum of daily mean temperatures exceeding 0°C over a one-year period to represent the annual accumulated temperature. Additionally, we have incorporated a concise explanation for this concept in the draft.

Q11:Could be useful to mention which graphical software was used to make figures 3-5.

A:Dear Professor, thank you for your advice, we have added the name of graphical software used to create figures 3-5 in the right situation in the draft.

Results

Q12:Line 202. Fig. 4. Should be better to delete “It’s a figure to show the” and simply start the text with “Spatial variation of……”.

A:Dear Professor, thank you for your advice, we have revised the description and start the text with “Spatial variation of……”.

Q13:Line 210, 211. Text already presented in Table 2, so decide which way to present Resistance and Habitat quality data, in a paragraph or a table. Rewrite the paragraph accordingly to the decision.

A:Dear Professor, thank you for your advice, we have deleted the text, and kept it in table 2. Furthermore, we have rewrote this paragraph.

Q14:Table 3. Use of percent or percentage? Correct.

A:Dear Professor, thank you for your advice, we decided to use percent in this table.

Q15:Table 4. Make shorter community characteristics column and expand the previous four columns for better reading.

A:Dear Professor, thank you for your advice, we have modified the form for better reading.

Discussion

Q16:Line 292. Tang does not match with number [54]. Correct.

A:Dear Professor, thank you for your advice, we have changed it to a right one.

Q17:Use italics for the scientific names.

A:Dear Professor, thank you for your advice, we have changed the scientific names of species in italics.

######

Response to reviewer 2:

Dear professor，

Thanks for your precious advice. These advice possess great significance for improving the quality of our manuscript. We attached great importance to these, and took actions carefully to revise our manuscript. In this response letter we attached the advice for our manuscript and made the response under it. Furthermore, we resubmitted the revised manuscript to submission system, and the texts we had revised were marked in red in the new one. The details of modification below as:

The manuscript PONE-D-24-44569 contains important findings that contribute to our understanding of spontaneous or natural" vegetation succession across different climatic conditions in Yunnan, China. The findings will be of significant interest to the broad scientific community and are backed by a strong data set spanning 30 years of observations.However, this does not come through from the abstract nor the introduction, and it is hard to understand from the current presentation of the methods.

Q1:In order to better understand and appreciate the findings and implications of this study a better organization of the methods is needed, which then can be used to re-write the abstract and discussion. For example, the abstract does not do justice to the span of sites, observations, and intervals of the observation used in this study.Only after reading halfway into the manuscript is enough information presented on the 30-year data set that has 6 observations taken at five-year intervals.

A:Dear Professor, thank you for your advice. For abstract, we have rewrote this section, which include the motivation of why we conducted this research, more information to present the 30-year data set, the categories of sub-area where came from, and the detailed information of result(natural succession index value, range of succession time in each sub-areas, range of stand volume in each sub-areas, relations between them). For methods, we have revised this section, which include more information to show how we divided the study area into sub-areas and details of crude data in this research part, more details to show how to calculate the natural succession index, and more information of observation system (observations, details of quadrat, intervals of observation, span of sites). For discussion, we have rewrote this section, include some unreasonable paragraph structures, some inconsistent sentences, and some vague expressions.

Q2:The authors are suggested to please improve readability of the text narrative by providing information on the study design and how locations-sites/plots/samples/time intervals. As it is currently presented it does not convey the importance and complexity of the results being present, and it even detracts from appreciating the implications from the findings.

A:Dear Professor, thank you for your advice. We have added the study design and how location locations-sites/plots/samples/time intervals in the Materials and monitor system section.

Q3:Please add to Table 1 a blank row to provide clarity and space - between the "Primary division" categories and for Table 4 adding a blank row to provide clarity and space - between the "Secondary division" categories.

A:Dear Professor, thank you for your advice. We have added enough space in every row in these table to improve the clarity and space between each categories.

Q4:The information of Figure 2 and Figure 3

A:Dear Professor, this advice is not a completed one. But we also checked these figure in our draft to avoid some obvious mistakes.

---

## [Decision Letter · Decision Letter 1]

3 Feb 2025

Using Natural Vegetation Succession to Evaluate How Natural Restoration Proceeds under Different Climate in Yunnan, Southwest China

PONE-D-24-44569R1

Dear Dr. Gui,

We’re pleased to inform you that your manuscript has been judged scientifically suitable for publication and will be formally accepted for publication once it meets all outstanding technical requirements.

Kind regards,

Ghulam Yaseen, Ph.D.

Academic Editor

PLOS ONE

Additional Editor Comments (optional):

Reviewers' comments:

Reviewer's Responses to Questions

**Comments to the Author**

1. If the authors have adequately addressed your comments raised in a previous round of review and you feel that this manuscript is now acceptable for publication, you may indicate that here to bypass the “Comments to the Author” section, enter your conflict of interest statement in the “Confidential to Editor” section, and submit your "Accept" recommendation.

Reviewer #1: All comments have been addressed

2. Is the manuscript technically sound, and do the data support the conclusions?

Reviewer #1: Yes

3. Has the statistical analysis been performed appropriately and rigorously? 

Reviewer #1: Yes

4. Have the authors made all data underlying the findings in their manuscript fully available?

Reviewer #1: Yes

5. Is the manuscript presented in an intelligible fashion and written in standard English?

Reviewer #1: Yes

6. Review Comments to the Author

Reviewer #1: The authors have addressed the majority of the recommendations and now I think the paper can reach a publication level. However it will be worthy if authors could make a final full read of the paper to improve language writing though is not their native language. All the rest of the topics I think can be already published as it stand.

7. PLOS authors have the option to publish the peer review history of their article (what does this mean? ). If published, this will include your full peer review and any attached files.

**Do you want your identity to be public for this peer review?** For information about this choice, including consent withdrawal, please see our Privacy Policy .

Reviewer #1: **Yes: ** Jose Luis Martinez S

---

## [Editor Report · Acceptance letter]

PONE-D-24-44569R1

PLOS ONE

Dear Dr. Gui,

I'm pleased to inform you that your manuscript has been deemed suitable for publication in PLOS ONE. Congratulations! Your manuscript is now being handed over to our production team.

Kind regards,

on behalf of

Professor Ghulam Yaseen

Academic Editor

PLOS ONE